# Changes in Textural Quality and Water Retention of Spiced Beef under Ultrasound-Assisted Sous-Vide Cooking and Its Possible Mechanisms

**DOI:** 10.3390/foods11152251

**Published:** 2022-07-28

**Authors:** Hengpeng Wang, Ziwu Gao, Xiuyun Guo, Sumin Gao, Danxuan Wu, Zongzhen Liu, Peng Wu, Zhicheng Xu, Xiaobo Zou, Xiangren Meng

**Affiliations:** 1Key Laboratory of Chinese Cuisine Intangible Cultural Heritage Technology Inheritance, Ministry of Culture and Tourism, College of Tourism and Culinary Science, Yangzhou University, Yangzhou 225127, China; yzuwhp@163.com (H.W.); gzw96530@163.com (Z.G.); 007840@yzu.edu.cn (X.G.); gsumin@163.com (S.G.); wudanxuan1998@163.com (D.W.); zzl4594559@163.com (Z.L.); wupeng4578@sina.com (P.W.); 13913835656@163.com (Z.X.); 2Agricultural Product Processing and Storage Lab, International Joint Research Laboratory of Intelligent Agriculture and Agriproducts Processing, School of Food and Biological Engineering, Jiangsu University, Zhenjiang 212013, China; zou_xiaobo@ujs.edu.cn

**Keywords:** ultrasound, sous-vide cooking, spiced beef, textural quality, water retention

## Abstract

The present study investigated the effects of ultrasound (28 kHz, 60 W at 71 °C for 37 min) combined with sous-vide cooking (at 71 °C for 40, 60, 80, 100, 120 min) on the textural quality, water distribution, and protein characteristics of spiced beef. Results showed that the spiced beef treated with conventional cooking (CT) had the highest cooking loss (41.31%), but the lowest value of shear force (8.13 N), hardness (55.66 N), springiness (3.98 mm), and chewiness (64.36 mJ) compared to ultrasound-assisted sous-vide (USV) and sous-vide cooking (SV) groups. Compared with long-time thermal treatment, USV heating within 100 min enhanced the water retention of spiced beef by maintaining the lower values of cooking loss (16.64~25.76%), *T*_2_ relaxation time (242.79~281.19 ms), and free water content (0.16~2.56%), as evident by the intact muscle fibers. Moreover, the USV group had relatively lower carbonyl content, but higher sulfhydryl content compared to CT and SV groups. More protein bands coupled with a minor transformation from α-helixes to β-turns and random coils occurred in USV40~USV80. In conclusion, these results indicated that USV treatment within 100 min positively affected the textural quality and water retention of spiced beef by moderate protein oxidation.

## 1. Introduction

Spiced beef is a traditional Chinese sauce pickled product made from beef as the primary raw material, seasoned with seasonings and spices, and then pre-cooked, soaked, and cooked. Consumers appreciate its delicious flavor and nutritional benefits [1]. However, the processing of spiced beef is facing some problems including backward production methods, high cooking loss, difficulty in maintaining flavor, and unsuitability for industrial production [2]. Thus, current research focuses on improving production efficiency while maintaining product quality and reducing cooking loss using appropriate processing methods.

Sous-vide cooking is a food processing method in which ingredients are vacuum-packed and then heated at a temperature of 50~80 °C for a long time [3]. Numerous studies have reported that sous-vide cooking could effectively reduce moisture loss and slow down the oxidation of meat during the cooking process [4,5]. Moreover, the heating parameters of sous-vide cooking could be precisely controlled to give consistent and repeatable results [6]. The possibility of food contamination during processing was reduced to a certain extent due to the vacuum and oxygen-free environment [7]. However, long-term heating would lead to higher energy consumption, which did not match the goal of increasing production efficiency. Therefore, how to improve the heating efficiency of sous-vide cooking is an essential issue in the meat industry.

Ultrasound in meat processing was extensively studied for curing, tenderization, and sterilization. It was proved effective in shortening the processing time and maintaining meat quality [8]. For example, Kang et al. [9] reported that ultrasound-assisted curing treatments could improve the curing efficiency and tenderness of beef. Zhang et al. [2] found that ultrasound-assisted cooking could improve the eating quality of marinated beef and inhibit the growth of the microorganisms during storage. Additionally, Pohlman et al. [10] proved that ultrasound treatment could significantly improve heat transfer efficiency and reduce the cooking loss of beef. Therefore, we speculated that combining ultrasound and sous-vide cooking is feasible to improve the heating efficiency and edible quality of meat products.

Unfortunately, studies rarely focus on spiced beef processed by ultrasound-assisted sous-vide cooking. In particular, the changes in textural quality and water retention of spiced beef under the synergistic effects of ultrasound and sous-vide cooking and their possible mechanisms are still indistinct. Therefore, the main objectives of this study were: (1) to assess the effect of ultrasound-assisted sous-vide cooking on the textural quality and water retention of spiced beef; (2) to evaluate the protein characteristics of spiced beef treated with ultrasound-assisted sous-vide cooking by measuring carbonyl, sulfhydryl content, and protein structure; and (3) to clarify the possible mechanisms of the physicochemical quality changes in spiced beef under the synergistic treatment by partial least squares discriminant analysis (PLS-DA). The results could provide a theoretical reference for processing high-quality spiced beef products.

## 2. Materials and Methods

### 2.1. Materials and Chemicals

Six male Chinese Simmentals (18 months old, the average weight of 600 ± 10 kg) were selected at random from Shandong Huasheng Halal Meat Co., Ltd. (Jining, China). All cattle were fed under the same rearing conditions and slaughtered following the same standard commercial procedures. After being slaughtered, the anterior tendons were divided from the cattle and aged for 72 h. Salt, sugar, cooking wine, soybean sauce, and spices were provided by Suguo Supermarket Co., Ltd. (Yangzhou, China). All chemicals were of at least analytical grade and bought from China Pharmaceutical Group Co., Ltd. (Beijing, China).

### 2.2. Preparation of Meat and Curing Solution

After reaching 72 h of ageing, the connective tissue and the visible fat were removed from the anterior tendons. Each anterior tendon was randomly divided into four pieces (8 cm × 6 cm × 3 cm) with a weight of 150 ± 5 g, and then vacuum-packed and stored at −40 °C.

The curing solution was prepared according to the methods of Zou et al. [11] with slight modifications. Based on the weight of meat, a condiment was prepared by mixing 90 g/kg soybean sauce, 150 g/kg cooking wine, 15 g/kg salt, and 50 g/kg sugar. The spices were prepared by mixing 25 g/kg spring onions, 25 g/kg ginger, 5 g/kg star anise, 5 g/kg cinnamon, 0.3 g/kg cloves, 2.5 g/kg Shannai, 1 g/kg cumin, 2.5 g/kg incense leaves, 2.1 g/kg grass fruit, and 1 g/kg licorice in a piece of gauze and wrapping them up. The above condiments and spices were added to the water to be heated. The power of the electromagnetic furnace (C22-HT2218, Midea Living Appliances Manufacturing Co., Ltd., Foshan, China) was first set to 1600 W, and the curing solution was maintained at a boil for 10 min. Then, the power was adjusted to 800 W, 50 min was maintained, and the power was adjusted to 300 W and maintained for 60 min. Finally, the boiled curing solution was cooled and stored at 4 °C.

### 2.3. Preparation of Spiced Beef

The processing of spiced beef under different thermal treatments is shown in Figure 1. All meat samples were thawed in the refrigerator (0~4 °C) for 24 h in advance, and then marinated for 10 h with 3 g/kg salt (based on meat weight). The marinated samples were randomly divided into seven processing treatment groups of six each (*n* = 42).

Ultrasound-assisted sous-vide cooking of spiced beef: The marinated samples were transferred to self-sealing bags and added to the curing solution (1:2, *m*/*v*). The plastic bags containing the muscles and the marinade were placed in a vacuum tumbler (HkS-30VT, Hakexun Industrial & Trading Co., Ltd., Wuxi, China) for 120 min (−0.08 MPa, 0~4 °C, 7 r/min) and then removed. The self-sealing bags were placed at 0~4 °C to allow the muscles to continue to soak for 24 h. After being removed from the self-sealing bags, the meat samples were evacuated in vacuum bags. The muscles in vacuum bags were randomly divided into five groups of six each (*n* = 30) and placed in a piece of ultrasonic-assisted cooking equipment (RC-1000LG, Renchuan Technology Co., Ltd., Langfang, China) (ultrasonic frequency: 28 kHz, ultrasonic power: 60 W, ultrasonic time: 37 min, cooking temperature: 71 °C). The spiced beef was cooked for 40, 60, 80, 100, and 120 min and then removed. The curing solution was poured out of the vacuum bag and recorded as USV40, USV60, USV80, USV100, and USV120.

Sous-vide cooking of spiced beef: Muscle pre-cooking treatment was consistent with ultrasound-assisted sous-vide cooking of spiced beef. The vacuum packaged samples (n = 6) were placed in a constant temperature water bath (HH-6, Boke Bioindustries Ltd., Jinan, China) for sous-vide cooking (cooking temperature: 71 °C, cooking time: 120 min). The sous-vide cooking of spiced beef was recorded as SV.

Conventional cooking methods of spiced beef: The marinated samples (n = 6) were added to the pot in cold water, heated to boiling, and maintained at boiling temperature for 5 min before being removed and washed in cold water. Subsequently, the muscles were placed in a pot containing the prepared curing solution for heating. The power of the electromagnetic furnace was first set to 1600 W, and the curing solution was boiled for 10 min. Then, the power was adjusted to 800 W, maintained for 50 min, and finally, the power was adjusted to 300 W and maintained for 60 min. The heating time of conventional spiced beef was controlled at 120 min and recorded as CT.

### 2.4. Meat Quality Testing

#### 2.4.1. Measurement of Cooking Loss

The cooking loss was detected as previously described by Xia et al. [12] with a slight modification. The surface of anterior tendons samples obtained from all muscles was dried and weighed as *W*_1_ after 24 h of thawing. After being cooked, the surfaces of the spiced beef samples were dried and weighed as *W*_2_. Cooking loss was calculated using the following formula:(1)$$\mathrm{Cooking}\;\mathrm{loss}\;(\%)=\frac{W_{1}-W_{2}}{W_{1}}\times 100$$

#### 2.4.2. Measurement of Warner–Bratzler Shear Force (WBSF)

The WBSF was assessed by referring to the method of Ge et al. [13]. Each spiced beef sample obtained from random meat blocks was cut into three long strips (1 cm × 1 cm × 3 cm) parallel to the direction of muscle fibers. The strip was cut perpendicular to the direction of muscle fibers using a digital display muscle tenderness meter with a 50-kg load sensor (C-LM_3B_, Yangzhou University, Yangzhou, China). The test speed was 250 mm/min. The maximum peak of shear force was recorded, and the WBSF was expressed in Newtons (N).

#### 2.4.3. Analysis of Texture Profile

The textural properties were determined according to the method of Wang et al. [14]. Approximately 5 g samples were cut from random spiced beef blocks (2 cm× 2 cm × 2 cm). The measurement was carried out by using a texture analyzer (TA-XT. plus, Stable Micro Systems, Godalming Surrey, UK) with a P/100 column probe, a pre-test speed of 60 mm/min, a test speed of 120 mm/min, a test deformation of 60%, and a trigger force of 0.4 N. Each spiced beef sample’s hardness, springiness, and chewiness were measured.

#### 2.4.4. Analysis of Microstructure by Light Microscopy

The spiced beef samples obtained from random meat blocks were cut into three cubes (2.5 mm × 2.5 mm × 2 mm), which were then fixed in 4% (*w*/*v*) tissue fixative at 4 °C for 12 h. The samples were then fixed and embedded in paraffin. Thin slices were then made from the paraffin samples. The slides were dewaxed, stained with hematoxylin and eosin, and sealed with a coverslip. The samples were observed with an optical 200× light electron microscope (IX73 IX71, Iroda Instruments & Equipment Co., Nanjing, China), and histological images were obtained for analysis.

#### 2.4.5. Low-Field Nuclear Magnetic Resonance (LF-NMR) Analysis

Water distribution was measured by the method of Wang et al. [15]. The 2D *T*_1_-*T*_2_ relaxation measurements were performed on an LF-NMR Analyzer (AccuFat-1050, Magmai Co., Ltd., Nanjing, China). Approximately 25 g samples were cut from spiced beef with different thermal treatments (2.5 cm × 5 cm × 1 cm) and placed in cylindrical glass tubes for NMR measurements. *T*_1_-*T*_2_ spectroscopy of the spiced beef samples, to determine relaxation times, was obtained by IR-CPMG (Infrared-Carr-Purcell-Meiboom-Gill) sequences. Four independent scan repetitions were conducted for one sample. Other parameter settings: receiving gain: 120, echo interval: 0.2 ms, number of samples: 3000, interval time: 2 s, operating temperature: 36 °C, spectrometer frequency: 10 MHz.

### 2.5. Protein Properties Testing

#### 2.5.1. Extraction of Total Protein

Approximately 2 g spiced beef samples of each treatment group were added with 8 mL 2% (*w*/*v*) sodium dodecyl sulfate (SDS) and homogenated in the ice bath (9500 r/min, 2 × 30 s; 13,500 r/min, 2 × 30 s). The mixture was centrifuged for 20 min (4000× *g*), and the supernatant was retained as total protein extraction. The concentration of proteins was measured by the Biuret method.

#### 2.5.2. Carbonyl Content Determination

The protein carbonyl content was measured by slightly modifying the method described by Zhang et al. [16]. Protein solution (200 μL, 2 mg/mL) was added to 1 mL of 20% (*w*/*v*) trichloroacetic acid (TCA). The above mixture was centrifuged at 4 °C (12,000× *g*, 5 min). After centrifugation, the protein precipitate was washed with TCA (1 mL, 10%, *w*/*v*) and centrifuged (12,000× *g*, 5 min). Then, the protein precipitate was mixed with 2 mL of 10 mM 2, 4-dinitrophenylhydrazine (dissolved in 2 M HCl) and incubated at 37 °C for 30 min. Then, 1 mL of 2 M HCl was added to the blank control tube, thoroughly vortexed, and reacted for 30 min at 37 °C away from light. The solution was centrifuged at 12,000× *g* for 5 min at 4 °C. Then, the supernatant was discarded, and 1 mL of TCA (20%, *w*/*v*) was added to precipitate the protein. After centrifugation, the precipitate was washed with 1 mL of ethanol and ethyl acetate (1:1, *v*/*v*) to remove unreacted DNPH (2,4-dinitrophenylhydrazine) until no yellowing was observed. The pellet was dissolved in 2 mL of guanidine hydrochloride (6 M, dissolved in 20 mM of PBS buffer at pH 7.0), kept at 4 °C for 12 h, and centrifuged at 12,000× *g* for 5 min. The supernatant absorbance was measured using a spectrophotometer at 370 nm (U-3900, Hitachi Corp., Tokyo, Japan). The sample mixed with 2 M HCl instead of 2,4-dinitrophenylhydrazine was used as a blank. The carbonyl content was calculated using an absorption coefficient of 22,000 M^−1^ cm^−1^ and expressed as nmol/mg protein.

#### 2.5.3. Sulfhydryl Content Determination

Free and total sulfhydryl (SH) contents were determined according to the method described by Kang et al. [17]. For free SH group content, 100 μL of protein solution (2 mg/mL) was added to 1 mL of Tris buffer A (50 mM Tris-HCl, 10 mM Ethylene diamine tetraacetic acid, 0.6 M KCl, pH 8.3), followed by 20 μL of 10 M 5, 5-dithiol-bis (2-nitrobenzoic acid) (dissolved in 100 mM Tris-HCl, pH 7.0). Then, the solution was placed at 25 °C for 60 min. For total SH content, 100 μL of protein solution (2 mg/mL) was added to 1 mL of Tris buffer B (50 mM Tris-HCl, 10 mM Ethylene Diamine Tetraacetic Acid, 0.6 M KCl, 8 M urea, pH 8.3), followed by 20 μL of 10 M 5,5-dithiol-bis(2-nitrobenzoic acid) (dissolved in 100 mM Tris-HCl, pH 7.0). Then, the solution was placed at 25 °C for 30 min. The mixture was centrifuged for 5 min (10,000× *g*), and the absorbance value in the supernatant was measured at 412 nm. The SH content was calculated using a molar extinction coefficient of 13,600 M^−1^ cm^−1^. The result was expressed as nmol/mg protein.

#### 2.5.4. Fourier Transform Infrared (FTIR) Spectroscopy Analysis

Protein secondary structure content was determined according to the method of Gangidi et al. [18] with slight modifications. The spiced beef samples were vacuum freeze-dried and ground, and then scanned by placing an appropriate amount of sample on the diamond ATR attachment (Cary 5000, Varian Co., Palo Alto, CA, USA). The parameters were set as follows: measuring range: 400~4000 cm^−1^, number of scans: 100, scanning rate: 0.63 cm/s, resolution: 32 cm^−1^. The acquired spectra were judged by evaluating the second derivative spectra, and the amide I band (1700~1600 cm^−1^) was used to analyze the secondary structure.

#### 2.5.5. Sodium Dodecyl Sulfate-Polyacrylamide Gel Electrophoresis (SDS-PAGE) Analysis

The protein solution (3 mg/mL) was mixed with a 5 × loading buffer with or without β-mercaptoethanol. The mixture was boiled at 100 °C for 10 min. Subsequently, 7 μL of a mixture or 3 μL of molecular standard marker (Thermo Fisher Scientific Co., Ltd., Shanghai, China) was loaded in each lane of the precast gel (12% polyacrylamide, 10 wells) (GenScript Co., Ltd., Piscataway, NJ, USA). Electrophoresis was performed using a Mini-Protean Tetra System (Bio-RadLaboratories, Hercules, CA, USA). The electrophoresis parameters were 80 V for 30 min, followed by 110 V for approximately 70 min at 4 °C. After separation, the gel was stained for 60 min with Coomassie Brilliant Blue (0.1%, *w*/*v*), and decolorized using a decolorizing solution (methanol (10%, *w*/*v*), acetic acid (10%, *w*/*v*; 1:1)) until the bands became clear.

### 2.6. Statistical Analysis

All measurements were independently replicated at least three times, and a completely randomized design was used. The results were expressed as the mean ± standard deviation. Statistical analysis was performed by ANOVA (Analysis of variance) using SPSS (Statistical Product Service Solutions) 19.0 (SPSS Inc., Chicago, IL, USA). Duncan’s multiple range test evaluated significant differences between groups when *p* < 0.05. Partial least squares discriminant analysis (PLS-DA) was performed using the SIMCA software (v.14.1, Sartorius Stedim Co., Ltd., Aubagne, France).

## 3. Results and Discussion

### 3.1. Changes in Cooking Loss and Warner–Bratzler Shear Force

As shown in Figure 2A, the CT treatment had the highest cooking loss (41.31%), followed by the SV, USV80, USV100, and USV120 groups. The lowest value of the cooking loss was found in the USV40 treatment (16.64%), but there were no significant differences (*p* > 0.05) in cooking loss between USV80, USV100, USV120, and SV groups, which indicated that ultrasound-assisted sous-vide treatment could better improve the water retention of spiced beef. As presented in Figure 2B, compared with CT treatment, a significant increase (*p* < 0.05) of shear force was observed in SV and USV treatment. The result could be attributed to collagen dissolving and myofibril destruction by continuous heating with higher temperatures in conventional cooking treatment [19,20]. USV80, USV100, USV120, and SV groups all had relatively higher values of shear force, which could be mainly caused by heat-induced changes in muscle structure and protein denaturation during heating [21].

### 3.2. Changes in Texture Properties

As shown in Table 1, the cooking time, USV treatment, and interaction significantly affected (*p* < 0.05) the hardness, springiness, and chewiness value of spiced beef. The samples under CT treatment had the lowest hardness value (55.66 N) due to the destruction of tissue structure by long-time heating with higher temperatures. The highest hardness was observed in the USV120 treatment (98.25 N), but had no significant difference (*p* > 0.05) compared to the SV treatment. Compared with USV120 and SV treatments, the hardness was significantly reduced (*p* < 0.05) in samples under the USV40~USV100 treatment. The result might be attributed to the damage of tissue cells and the leakage of intracellular compounds due to the cavitation effect of ultrasound [22]. As presented, the effect of ultrasound-assisted sous-vide cooking on the hardness depends on the heating time, so by increasing the time from 40 to 120 min, the hardness increased.

According to Table 1, the springiness and chewiness value of spiced beef in USV treatment showed an increasing trend with the extension of heating time. The spiced beef heated for 80~120 min with USV treatment had a significantly higher value (*p* < 0.05) of springiness and chewiness than that in CT and SV groups, which was related to the moisture loss in the tissue that increased the elasticity of proteins by the relatively long-time heating with USV treatment [23]. Liu et al. [24] found that the water-holding capacity was significantly correlated with the springiness of meat. The changes in the texture of spiced beef were consistent with the cooking loss. As a result, the low-intensity ultrasound combined with low-temperature and short-time heating could reduce the damage to myofibrillar structures and alleviate the chewy and hard texture caused by heating alone [25,26]. Our findings suggested that USV treatment exhibited a notable improvement in maintaining the textural properties of spiced beef, and the heating time is recommended to be controlled at 80~100 min.

### 3.3. Changes in Light Microscopy

As shown in Figure 3, in CT treatment, the fiber membranes were damaged to a higher degree due to the long-time heating with higher temperature, as evident by the blurred borders of the adjacent muscle fibers, which explained the mechanism of the higher cooking loss and loose texture of spiced beef under conventional treatment. Compared with CT and SV treatment, the gap between muscle fibers was enlarged in the samples treated with USV. The result might be ascribed to the cavitation effect of ultrasound by expanding myofibrillar internal space to retain more water [27,28], which corresponded to our findings in cooking loss (Figure 2A). Moreover, the USV40 and USV60 groups had neatly aligned muscle fibers and larger diameters. Nevertheless, the extension of cooking time decreased the muscle fiber diameter but increased the fiber gap. Notably, the muscle fibers of spiced beef under USV treatment remained intact throughout the cooking process, which further confirmed that the higher water-holding capacity and texture quality of spiced beef could be maintained by USV.

### 3.4. Changes in Water Migration and Distribution

Low-field nuclear magnetic resonance (LF-NMR) is a rapid and non-destructive technique for determining proton relaxation properties associated with water status in muscles [29]. Previous studies on the moisture state of meat products mainly focused on *T*_2_ relaxation [30], while applying the two-dimensional (2D) LF-NMR *T*_1_-*T*_2_ relaxation spectroscopy technique to spiced beef was rarely reported. The 2D LF-NMR spectroscopy technique could collect all IR and CPMG sequence parameters and obtain both transverse and longitudinal relaxation properties [31]. As shown in Figure 4, the *T*_1_ and *T*_2_ relaxation NMR spectra were presented in the left and upper parts of the figure, respectively. Three central populations could be observed in the 2D *T*_1_-*T*_2_ relaxation NMR spectra, which referred to three forms of water in spiced beef. From left to right, these represented bound water (*T*_2b_, in the range of 0.1~10 ms), immobilized water (*T*_21_, in the range of 10~100 ms), and free water (*T*_22_, in the range of 100~1000 ms) [32,33].

The changes in relaxation time and their peak areas for different states of moisture in spiced beef are shown in Table 2.

With the extension of cooking time, *T*_1_ relaxation time increased gradually in USV groups, but it was significantly lower (*p* < 0.05) than in CT and SV groups. The prolongation of *T*_1_ relaxation time meant more free water migrated from the muscle, which might be attributed to the weakening water–protein interaction caused by long-time heating [34,35]. *T*_2_ relaxation time significantly decreased (*p* < 0.05) in USV groups heating for 40~80 min, but significantly increased (*p* < 0.05) at 100~120 min of heating. The result was related to the improvement of water retention capacity in spiced beef due to the cavitation effect of ultrasound, whereas the continued heating destroyed the integrity of myofibrils and accelerated the movement of internal moisture out of muscle tissue [36,37]. Moreover, the *T*_2_ relaxation time of spiced beef under CT treatment was significantly higher (*p* < 0.05) than that in SV and USV, which was mainly attributed to the severe denaturation of protein and damage to muscle fiber meshwork by long-time heating with higher temperature, which ultimately caused an obvious water loss between muscle cells [38,39]. The result was in accordance with our findings on cooking loss and the microstructure of spiced beef.

As presented, the content of immobilized water represented by *P*_2_ accounted for the largest proportion of the total water forms, reaching up to over 95% in samples treated with USV heating for 60 and 80 min. *P*_3_ represented the free water content in spiced beef and reached the highest value of 8.16% in CT treatment. These results further indicated that appropriate cooking time of USV treatment could obtain better water-holding capacity for meat, whereas long-time heating by CT and SV treatment was not conducive to improving water retention.

### 3.5. Changes in Protein Oxidation

The carbonyl and sulfhydryl (SH) content are considered as important indicators of protein oxidation [40]. As shown in Figure 5A, the carbonyl content in CT treatment was significantly higher (*p* < 0.05) than that in SV and USV groups, which might be attributed to the rapid moisture loss and destruction of the protein–water complex structure by long-time heating with higher temperature [41]. For the USV group, the carbonyl content gradually increased with prolonged cooking time, and leveled off at 100~120 min of cooking. In addition, the carbonyl content in the USV group was significantly lower (*p* < 0.05) than that in the SV treatment. This phenomenon was related to the encapsulation of some protein oxidation sites and peptide chains resulting from the cavitation effect of ultrasound, and the short-time heating with low-temperature was insufficient to expose those sites entirely [26]. The result was similar to the previous report by Yin et al. [42].

Sulfhydryl (SH) groups are easily oxidized to form disulfide bonds and a variety of dioxides during meat processing, ultimately resulting in a reduction of sulfhydryl content [43]. As shown in Figure 5B, the total and free SH content in the CT group was significantly lower (*p* < 0.05) than that in USV and SV treatment, which was consistent with the result of carbonyl content. The free SH content in USV samples significantly decreased (*p* < 0.05) during heating, suggesting the increment of protein oxidation levels by prolonged cooking time. Notably, the total SH content in the SV group was significantly lower (*p* < 0.05) than that in USV40, USV60, and USV80 samples, but had no significant difference (*p* > 0.05) with USV120 samples. The result indicated that the synergistic effect of prolonged low-temperature heating and ultrasound could further deepen oxidation levels by altering the protein structure [44,45].

### 3.6. Changes in Secondary Structure

As shown in Figure 6, the total absorbance of protein amino acid residues and the peak shapes were significantly changed with the prolonged cooking time. The amide I band (1600~1700 cm^−1^) is caused by the C=O, C−N stretching vibration, and the Cα−C−N, N−N in-plane bending vibration, which is the main spectral band reflecting the secondary structure of proteins [46]. Protein denaturation could be expressed as the transition of the α-helix (1646~1664 cm^−1^) and β-sheet (1615~1637 cm^−1^, 1682~1700 cm^−1^) to β-turns (1664~1681 cm^−1^) and random coils (1637~1645 cm^−1^) due to the increase of hydrophobicity and the destruction of hydrogen bonds [47,48].

The α-helix content decreased significantly (*p* < 0.05) with prolonged heating in USV samples. Simultaneously, the β-sheet content decreased to the minimum value of 31.67% when cooked for 100 min (Table 3). The α-helix is primarily an ordered arrangement within protein molecules that is stabilized by intramolecular hydrogen bonds, while the β-sheet is an ordered arrangement between proteins maintained by intermolecular hydrogen bonds [49]. The above results were mainly related to the weakening of hydrogen bonds due to the synergistic treatment of ultrasound and long-time heating. Notably, the β-turn and random coil content in USV100 and USV120 samples had no significant difference (*p* > 0.05) with CT and SV groups, which might be attributed to the changes in isoelectric point and hydrophobicity, caused by protein oxidation and degradation resulting from long-time heating [50].

Changes in protein secondary structure were significantly associated with water distribution in meat [51]. Previous studies also revealed a high correlation between cooking loss and α-helix structures [51,52]. Han et al. [29] suggested that the protein denaturation and aggregation could convert α-helixes and β-sheets to β-turns and random coils during heating. Our results further confirmed the relationship between water status and the secondary structure of meat protein, and explained the reason for poor water-holding capacity and higher protein oxidation levels occurring in spiced beef treated with conventional and individual sous-vide cooking.

### 3.7. Proteolytic Changes

The protein profiles performed by nonreduced and reduced SDS-PAGE from spiced beef under different thermal treatments were presented in Figure 7. The intensity of the myosin heavy chain (MHC) band in USV lanes increased compared with CT and SV, but decreased significantly with prolonged cooking time. As shown in Figure 7B, most of the reduced MHC and actin bands in USV lanes were restored under reducing conditions, while those in CT and SV lanes did not fully recover, which indicated that the proteins in CT and SV samples had been cross-linked as disulfide bonds and as non-disulfide bonds [53]. Kang et al. [9] found that the disulfide bond was mainly responsible for forming the MHC polymer after prolonged treatment.

The degradation of protein molecules was mainly characterized by blurring, weakening, and expanding bands at higher molecular weights, while lower molecular weight regions showed new bands or the deepening of band color [54]. In the present study, the USV60~USV120 lanes bands showed deepening at 10~34 kDa, blurring at 43~95 kDa, and slight diffusion at MHC. The result might be attributed to the oxidative degradation of meat protein by continuous heating. Notably, most protein bands in CT lanes disappeared, but they significantly strengthened in USV lanes, further indicating the apparent protein degradation in spiced beef by conventional cooking. In contrast, USV treatment could decrease the degrees of protein degradation and aggregation, which was consistent with the result of protein oxidation.

### 3.8. Partial Least Squares Discriminant Analysis (PLS-DA)

PLS-DA was carried out by using the protein characteristics as the independent variable, and the apparent quality indicators, such as textural quality and water distribution, as the dependent variables (Figure 8). A total of two principal components (Factor 1 and Factor 2) were extracted and represented 94.80% of the information on the original variables. As presented, the thermal treatment affecting independent and dependent variables in the PLS-DA plot was roughly divided into four circle areas. The SV, USV100, USV120, *P*_1_, β−turn, and random coil were distributed in the first quadrant with a significant correlation, which indicated that the relatively long-time heating by SV and USV treatment changed the protein conformation to a greater extent, resulting in the migration of bound water in spiced beef. Total sulfhydryl, free sulfhydryl, shear force, hardness, springiness, and chewiness were distributed in the second quadrant, suggesting that protein oxidation had a visible effect on the texture properties of spiced beef. The USV40, USV60, USV80, α−helix, β−sheet, and *P*_2_ were concentrated in the third quadrant, indicating that the USV treatment with relatively short time heating had a significant effect on the state of immobile water due to the changes of the α−helix and β−sheet in spiced beef. Furthermore, the CT treatment, cooking loss, carbonyl, *T*_1_, *T*_2_, and *P*_3_ were located in the fourth quadrant, which further confirmed that the protein oxidation had an essential effect on the water retention of spiced beef.

Notably, the plotting points representing different thermal treatments moved from left to right with the increase of cooking time and core temperature, suggesting that prolonged high-temperature heating caused the changes in protein conformation due to the excessive oxidation, and ultimately damaged the physicochemical qualities of spiced beef. However, adopting the USV treatment within 100 min could result in the moderate oxidation of proteins, which positively affected the texture properties and water retention of spiced beef.

## 4. Conclusions

This paper found that ultrasound-assisted sous-vide cooking could significantly reduce the cooking loss and improve the physicochemical qualities of spiced beef. The shear force, hardness, chewiness, springiness, and water retention of spiced beef under USV heating within 100 min were enhanced by the cavitation effect of ultrasound that swelled the myogenic fibers in beef. Additionally, USV treatment could significantly reduce the exposure of sulfhydryl and carbonyl groups by increasing the folding of protein structure. Furthermore, the results of SDS-PAGE, secondary structure, and PLS-DA further indicated that the USV treatment within 100 min could decrease the levels of protein degradation and aggregation, which positively affected the textural quality and water retention of spiced beef. Overall, ultrasound-assisted sous-vide cooking is an efficient and healthy method for meat processing. We recommend heating for 80~100 min by USV as the most suitable treatment for high-quality spiced beef.

## Figures and Tables

**Figure 1 foods-11-02251-f001:**
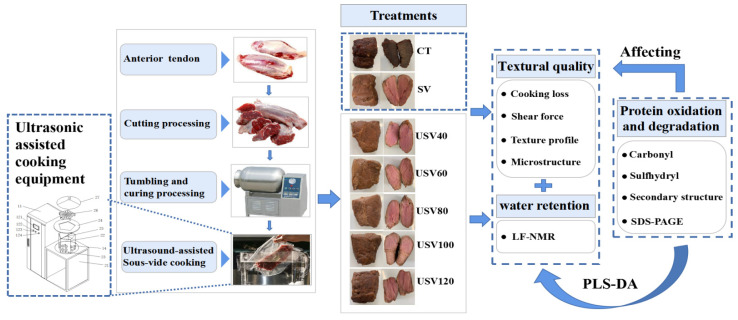
Flow chart of processing of spiced beef under different thermal treatments. CT: conventional cooking methods of spiced beef; SV: sous-vide cooking of spiced beef; USV40~USV120: ultrasound-assisted sous-vide cooking of spiced beef for 40~120 min.

**Figure 2 foods-11-02251-f002:**
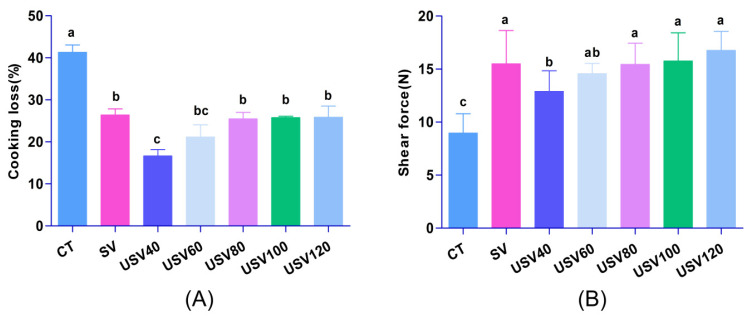
Changes in cooking loss (**A**) and shear force (**B**) of spiced beef under different thermal treatments. CT: conventional cooking of spiced beef; SV: sous-vide cooking of spiced beef; USV40~USV120: ultrasound-assisted sous-vide cooking of spiced beef for 40~120 min. Different lowercase letters indicate statistically significant difference at (*p* < 0.05).

**Figure 3 foods-11-02251-f003:**
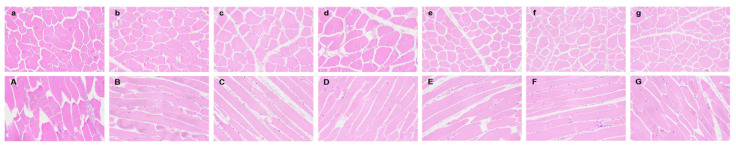
Light microscopy images of spiced beef under different thermal treatments at 200× magnification. (**a**–**g**) Transverse sections of myofibrils of the CT, SV, USV40, USV60, USV80, USV100, and USV120 treatment, respectively; (**A**–**G**) Longitudinal sections of myofibrils of the CT, SV, USV40, USV60, USV80, USV100, and USV120 treatment, respectively.

**Figure 4 foods-11-02251-f004:**
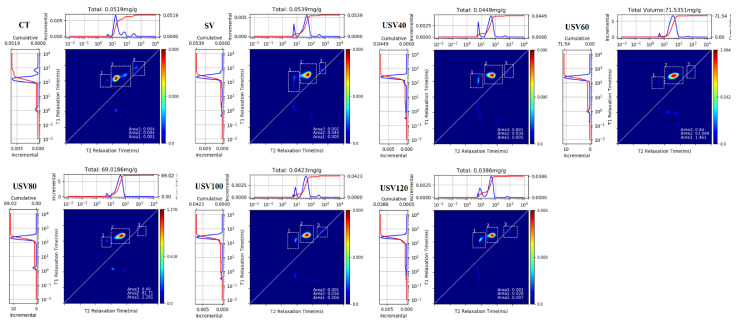
2D *T*_1_-*T*_2_ relaxation profiles of spiced beef under different thermal treatments.

**Figure 5 foods-11-02251-f005:**
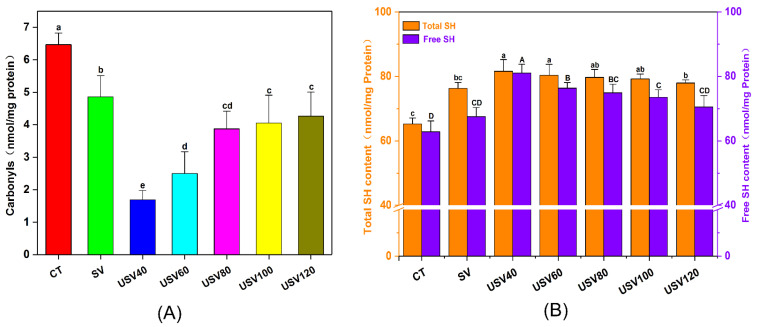
Changes in the carbonyl content (**A**) and sulfhydryl (SH) content (**B**) of spiced beef under different thermal treatments. Different letters (a~d and A~D) indicate a significant difference (*p* < 0.05).

**Figure 6 foods-11-02251-f006:**
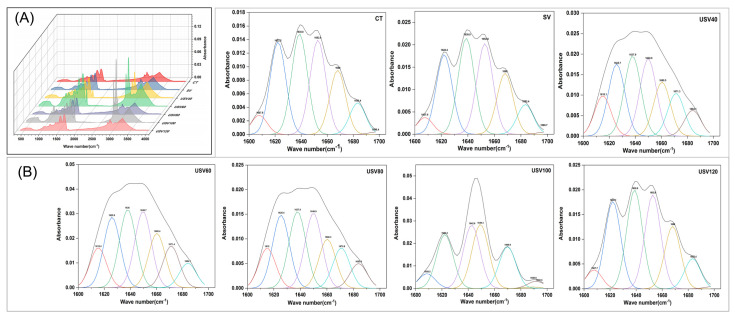
Infrared spectra of spiced beef proteins and fitted curve of protein amide I band of spiced beef under different thermal treatments. (**A**) Infrared spectra of spiced beef proteins; (**B**) fitted curve of protein amide I band of spiced beef.

**Figure 7 foods-11-02251-f007:**
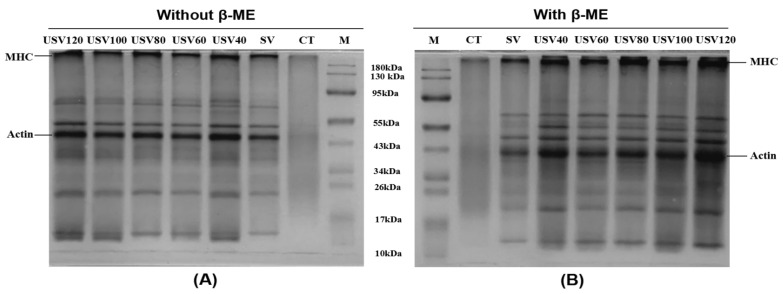
SDS-PAGE profiles of proteins in spiced beef under different thermal treatments. Samples were prepared without (**A**) and with (**B**) β-mercaptoethanol (β-ME) addition. M: protein marker.

**Figure 8 foods-11-02251-f008:**
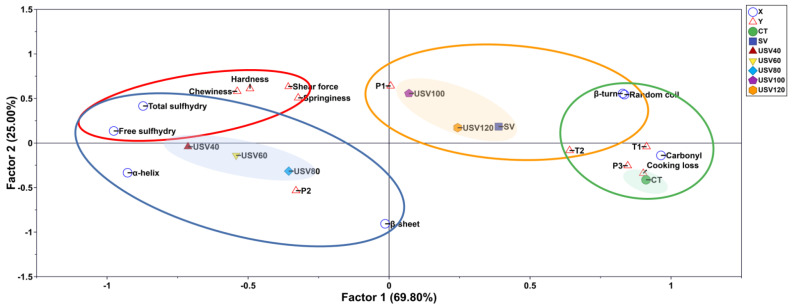
The correlation distribution of textural quality, water distribution, and protein characteristics of spiced beef under different thermal treatments. X: protein characteristics; Y: apparent quality indicators.

**Table 1 foods-11-02251-t001:** Effect of different thermal treatments on the textural properties of spiced beef.

Sample ID	Hardness (N)	Springiness (mm)	Chewiness (mJ)
CT	55.66 ± 4.48 ^c^	3.98 ± 0.64 ^c^	64.36 ± 9.50 ^c^
SV	96.34 ± 5.58 ^a^	4.59 ± 0.34 ^b^	210.80 ± 11.85 ^ab^
USV40	85.91 ± 4.36 ^bc^	4.55 ± 0.25 ^b^	191.37 ± 13.65 ^b^
USV60	92.62 ± 5.60 ^b^	4.60 ± 0.37 ^b^	203.04 ± 18.70 ^b^
USV80	92.19 ± 1.19 ^b^	4.98 ± 0.22 ^ab^	222.98 ± 13.27 ^a^
USV100	93.57 ± 2.70 ^b^	4.90 ± 0.16 ^ab^	218.65 ± 16.80 ^a^
USV120	98.25 ± 4.05 ^a^	5.33 ± 0.55 ^a^	246.27 ± 16.25 ^a^

CT: conventional cooking spiced beef; SV: sous-vide cooking of spiced beef; USV40~USV120: ultrasound-assisted sous-vide cooking of spiced beef for 40~120 min. Different letters in the same column indicate statistically significant difference (*p* < 0.05).

**Table 2 foods-11-02251-t002:** Effect of different thermal treatments on LF-NMR relaxation time and peak areas of water forms in spiced beef.

Sample ID	*T*_1_ (ms)	*T*_2_ (ms)	*P*_1_ (%)	*P*_2_ (%)	*P*_3_ (%)
CT	666.03 ± 18.59 ^a^	315.06 ± 10.58 ^a^	2.04 ± 0.04 ^d^	89.80 ± 2.14 ^b^	8.16 ± 0.55 ^a^
SV	609.70 ± 20.14 ^ab^	300.54 ± 10.21 ^ab^	9.80 ± 0.53 ^c^	86.27 ± 2.21 ^bc^	3.93 ± 0.68 ^b^
USV40	347.05 ± 12.50 ^d^	281.19 ± 9.56 ^b^	11.90 ± 0.75 ^b^	85.71 ± 3.02 ^c^	2.39 ± 0.84 ^c^
USV60	397.94 ± 10.53 ^c^	280.87 ± 8.74 ^b^	2.24 ± 0.25 ^d^	96.47 ± 4.05 ^a^	1.29 ± 0.15 ^cd^
USV80	386.15 ± 11.27 ^c^	242.79 ± 7.55 ^c^	3.54 ± 0.37 ^d^	95.70 ± 3.57 ^a^	0.76 ± 0.10 ^d^
USV100	488.85 ± 10.83 ^b^	261.27 ± 6.39 ^bc^	10.26 ± 0.90 ^b^	87.18 ± 2.56 ^bc^	2.56 ± 0.24 ^c^
USV120	424.34 ± 15.56 ^bc^	308.88 ± 7.87 ^ab^	19.44 ± 0.27 ^a^	77.78 ± 2.30 ^d^	2.78 ± 0.50 ^c^

Different letters in the same column indicate statistically significant difference (*p* < 0.05).

**Table 3 foods-11-02251-t003:** Changes in relative protein secondary structure content of spiced beef under different thermal treatments.

Sample ID	Secondary Structure Content (%)			
	α-Helix	β-Sheet	β-Turn	Random Coil
CT	23.35 ± 0.07 ^c^	36.01 ± 0.02 ^a^	15.81 ± 0.04 ^a^	24.83 ± 0.05 ^a^
SV	24.09 ± 0.08 ^b^	34.25 ± 0.10 ^b^	16.20 ± 0.03 ^a^	25.46 ± 0.03 ^a^
USV40	33.42 ± 0.04 ^a^	35.14 ± 0.15 ^b^	10.94 ± 0.08 ^b^	20.50 ± 0.08 ^b^
USV60	33.78 ± 0.14 ^a^	34.94 ± 0.03 ^b^	10.91 ± 0.13 ^b^	20.37 ± 0.13 ^b^
USV80	32.75 ± 0.05 ^ab^	36.26 ± 0.09 ^a^	10.68 ± 0.07 ^b^	20.31 ± 0.07 ^b^
USV100	25.86 ± 0.15 ^b^	31.67 ± 0.11 ^c^	16.87 ± 0.04 ^a^	25.59 ± 0.04 ^a^
USV120	23.89 ± 0.13 ^c^	35.21 ± 0.21 ^b^	15.85 ± 0.07 ^a^	25.05 ± 0.07 ^a^

Different letters in the same column indicate statistically significant difference (*p* < 0.05).

## Data Availability

The data presented in this study are available on request from the corresponding author.

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
