# Peer review of "Changes in Textural Quality and Water Retention of Spiced Beef under Ultrasound-Assisted Sous-Vide Cooking and Its Possible Mechanisms"

_foods, 2022, doi:10.3390/foods11152251_

Round 1

Reviewer 1 Report

Sous-vide cooking has been more and more popular technology and has many advantages over the conventional cooking. Ultrasound assisted processes are widely used in food industry for example to improve the textural properties, sensory quality or modify the enzymatic and microbial processes. Therefore, the topic of the manuscript can be considered as interesting for the readers. The mnauscript is general well structured. Introduction summarizes well the relevance of the study. Applied methods are adequate and described in enough details. Figures and tables represent well the results. Experimental results are discussed in details with relevant references.

Comments, suggestions:

In the Introduction section, the applicability and efficiency of of ultrasound-assisted processes are poorly discussed. Please improve this part. (line 58-66).

Please check the typos in the manuscript. (See ’ 3):’ in line 76, ten h’ in line 113,’for instance).

It is not clear how was determined the time of sonication.

The visibility of Figure 4 is poor. Please improve it.

Have the authors information related to the colour and or/sensory properties of different treated beef.

Reviewer 2 Report

The authors presented the effect of ultrasound combined with different sous-vide cooking time on the textural quality, water distribution, and protein characteristics of spiced beef. The idea of the article is good. Appropriate analyses are measured and good discussions are provided. However, some ambiguities are mentioned below that need to be cleared. A few points to improve the current format of the article will be mentioned below, too:

Minor editing of English language and style required. For example in abstract: “combining” should be change to “combined”; “comparing” should be change to “compared”; etc.

The abstract should be more informative by giving real results rather than elastic sentences. Important and main contents should be given. Support the results with some quantitative data. Moreover, no conclusions are provided.

How many samples were there per muscle?

The design is not clear– were the samples treated as one group per treatment, the level of replication is not clear.

Why didn't you measure the pH? To analyze many results, knowing the pH is essential.

You need to include ALL the model terms. The following article may be helpful - Biffin TE, et al. 2020. The effect of whole carcase medium voltage electrical stimulation, tenderstretching and longissimus infusion with actinidin on alpaca quality. Meat Sci. 164 Article 108107. All fixed and random effects need to be included for the analysis of all traits. Line 120 implies that the experiment was replicated, thus this needs to be included as a random term in the model?  You could also have samples within muscles as a random term, and muscle as a random term.

Regarding the investigation of the effect of ultrasound on the textural characteristics of meat, it is good to use the following article (The combined effect of ultrasound treatment and leek (Allium ampeloprasum) extract on the quality properties of beef) in the analysis of your results.

Conclusion: what is the future of your findings? Conclusion is not insightful, what are suggestions?

Round 2

Reviewer 2 Report

Almost all the mentioned points have been modified by the authors.

Author Response

Thanks so much for your careful review and good suggestion on our manuscript, It helps us improve a lot.